# Manufacture and Deformation Angle Control of a Two-Direction Soft Actuator Integrated with SMAs

**DOI:** 10.3390/ma17030758

**Published:** 2024-02-05

**Authors:** Aline Iobana Acevedo-Velazquez, Zhenbi Wang, Anja Winkler, Niels Modler, Klaus Röbenack

**Affiliations:** 1Institute of Control Theory, Dresden University of Technology, 01062 Dresden, Germany; klaus.roebenack@tu-dresden.de; 2Institute of Lightweight Engineering and Polymer Technology, Dresden University of Technology, 01307 Dresden, Germany; zhenbi.wang@tu-dresden.de (Z.W.); anja.winkler@tu-dresden.de (A.W.); niels.modler@tu-dresden.de (N.M.)

**Keywords:** shape memory alloys, soft actuator, 3D printed soft structures, thermoplastic polyurethane, system identification, PI controller, closed-loop control

## Abstract

In this contribution, the development of a 3D-printed soft actuator integrated with shape memory alloys (SMA) wires capable of bending in two directions is presented. This work discusses the design, manufacturing, modeling, simulation, and feedback control of the actuator. The SMA wires are encased in Polytetrafluoroethylene (PTFE) tubes and then integrated into the 3D-printed matrix made of thermoplastic polyurethane (TPU). To measure and control the deformation angle of the soft actuator, a computer vision system was implemented. Based on the experimental results, a mathematical model was developed using the system identification method and simulated to describe the dynamics of the actuator, contributing to the design of a controller. However, achieving precise control of the deformation angle in systems actuated by SMA wires is challenging due to their inherent nonlinearities and hysteretic behavior. A proportional-integral (PI) controller was designed to address this challenge, and its effectiveness was validated through real experiments.

## 1. Introduction

The significance of smart materials has grown due to their capability to modify their characteristics in response to external stimuli. These materials can function both as sensors and actuators, making them highly versatile for use in driving, control, or support of flexible structures [1]. Specifically, Shape Memory Alloys (SMAs) are metallic materials that can return to a predefined shape or size when subjected to specific stimuli, typically temperature changes. They possess unique properties, allowing deformation at one temperature and recovering their original shape at another. SMAs have a superior force-to-weight ratio compared to other smart materials, making them suitable for fields such as aerospace, biomedicine, and robotics [2,3,4,5].

There are mainly two approaches for manufacturing SMA wire-driven compliant active parts found in the literature, they are post-mounting and integrating. Studies referenced in [6,7,8,9] involved producing passive compliant components first and then post-mounting SMA wires onto them. This method is not necessary to take into account the shortening of the SMA wires due to temperature activation during the production process of the passive components. However, its size and shape design are limited by the need to reserve space in the passive element for mounting. The integrating manufacturing approach can be understood as manufacturing composite materials with SMA wires in them. This approach eliminates the need for subsequent installation of SMA wires, enabling variant actuator movements by adjusting various component configurations within the composite material. Moreover, sensors can also be integrated with SMA wire together into composite materials to implement a controller [10].

Lohse et al. [11] knitted SMA wires into a glass fiber woven and injected silicon into it with vacuum assistance. Mersch et al. [12] braided copper and polyamide yarns around the SMA wire to form a core-sheath structure, stitched it in a certain shape onto a glass fiber fabric using tailor fiber placement (TFP) technology and injection molded it with silicon. The copper wires aim to sense the temperature of the SMA wire. They introduced approaches by integrating SMA wires into a soft matrix using resin injection. This approach has the advantage of a wide range of commercially available soft elastomer materials that can withstand high temperatures and are more easily compatible with high-temperature activated SMA wires, but the need for mold production and a multi-step process increases manufacturing costs.

Wang et al. [13] employed 3D printing technology to fabricate an SMA wire-driven compliant actuator, they designed a cavity within the actuator matrix for embedding the SMA wire during the 3D printing process of the matrix. The experimental results highlighted the feasibility and reliability of this manufacturing method. In comparison to the conventional resin injection process, 3D printing technology allows for complex part shapes and high resolution, as well as enables customized design and rapid prototyping. Moreover, the weight and stiffness distribution of 3D-printed components can be efficiently regulated by adjusting the infill percentage and modifying the infill structure, which is not possible with traditional injection techniques. Given the above advantages, a similar approach to [10] was adopted in this study to fabricate a soft actuator by utilizing 3D printing.

The optimal performance of an actuator driven by SMA relies heavily on precise modeling and control. Various modeling approaches contribute to the accurate representation of nonlinear relationships. Fischer et al. [14] used the statistical method of polynomial regression to model the self-sensing of SMA wire by fitting a polynomial equation to the measured data. Modeling with SMA constitutive models like the Tanaka model [15], Liang and Rogers model [16], and Brinson model [17], as well as hysteresis models such as the Preisach model [18], are found in the literature. Beyond modeling SMA, considering the entire system is crucial, an alternative approach uses experimental modeling also known as system identification to derive simpler equations, as demonstrated in [19,20], representing the nonlinear relationship with unstructured uncertainties models using a rational transfer function.

However, controlling the precise behavior of SMA-driven actuators is still challenging due to inherent hysteresis and nonlinearities. Various control approaches have been explored, including the application of an extended Kalman filter for a shape memory alloy arm [21], virtual reference feedback [22], active disturbance rejection control for flexible bioinspired robotics devices [23], and sliding modes such as model-free adaptive sliding mode, PID, MFAC for SMA-actuated systems [24]. Proportional Integral (PI) controllers, known for their ease of implementation, have been successfully applied, as seen in a robust PI controller for an interactive rubber composite [20] and a PI controller designed and validated for an SMA wire in experiments [25], aiming for optimal tuning.

The objective of this study is to develop an SMA wire driven compliant actuator prototype that can bend in two directions and be controlled to achieve or maintain a desired deformation angle. The matrix of the actuator prototype was designed as a flexible structure and 3D printed using thermoplastic polyurethane (TPU) filament material. Two SMA wires are encased in PTFE tubes and embedded symmetrically on both sides outside the neutral plane of the matrix during 3D printing, allowing for bi-directional bending. To address the control challenge, a robust control technique has been implemented, ensuring the achievement of specific deformations.

This paper is structured into seven sections. Section 2 outlines the manufacturing process, while Section 3 explores the experimental setup of the computer vision system employed for deformation angle measurement. Section 4 presents the development of a mathematical model to describe the dynamic behavior of the actuator using system identification method. Section 5 concentrates on the control strategy for achieving the desired deformation angle. In Section 6, we discuss the experimental results obtained by implementing the control. Lastly, Section 7 presents the conclusions.

## 2. Actuator Manufacture

The SMA wires utilized in this study are Nitinol SMA wires (SmartFlex® 300 μm) with a diameter of 0.3 mm, manufactured by SAES Getters (20045 Lenarte, Milan, Italy), and pre-stretched to 4–5% during production. The actuator matrix was 3D printed using thermoplastic polyurethane (TPU Flex Semisoft) with a hardness of A88, supplied by Filamentworld (89231 Neu-Ulm, Germany).

PTFE tubes with an inner diameter of 0.5 mm and an outer diameter of 0.76 mm were wrapped around the two SMA wires to form core-sheath structures that were then embedded into the actuator matrix during the 3D printing process. PTFE is a synthetic fluoropolymer of tetrafluoroethylene. This material is versatile and popular for its unique properties such as a low coefficient of friction and a wide operation range of temperatures. It can withstand extremely low to high temperatures without losing its properties [26].

In Figure 1 shows the manufacturing process of the actuator. The low friction properties of the PTFE enable the SMA wires to slide softly, ensuring that the shortening of the SMA does not disrupt the structure of the TPU matrix. In order to accommodate the SMA wires encased in PTFE tubes and to fix their shape in the matrix, two cavities were introduced into the matrix. A post-processing script called ’pause at height’ was employed in the slicing software CURA, causing a temporary pause in the 3D printing process at the second last layer of the cavity. This pause allowed for the sequential embedding of the SMA wires before the printing resumed. Figure 1 (left) shows different stages of the manufacturing and Figure 1 (right) provides the dimensions of the specimen.

The driving behavior of a SMA wire is related to the stretch force applied to it. To make the experimental results reproducible, a test bench as shown in Figure 2 was fabricated for applying a certain pre-stretch force to the two wires by two springs and fixing the actuator properly. The two ends of the springs are connected to two 3D-printed sliders respectively. The upper slider is fixed when the spring is stretched to a certain length, which determines the pre-stretch force subjected to the SMA wires. The spring can also provide some elastic potential energy to protect the correlative SMA wire and matrix material from passive stretching damage when activating the other SMA wire.

## 3. Experimental Setup for the Computer Vision System

To measure the deformation angle, an experimental computer vision setup was implemented. This setup is formed by a power supply, a light ring, an Intel RealSense camera, a computer, a driver circuit, and an Arduino UNO, as illustrated in Figure 3.

The deformation angle was determined using computer vision technology, involving three distinct reference points of varying colors. Specifically, blue and green points were placed on the actuator, while a red point was fixed on the test bench. Notably, the mobile point is the green one. Figure 4 shows the actuator prototype with measurement points on it on the left and a schematic diagram of the actuator with the variables used for calculating the deformation angle (denoted as α) on the right.

To compute the deformation angle using the camera, firstly the circular areas colored in red, green, and blue are identified. Subsequently, the centroid is calculated of each area given by the coordinate of the points $P_{R}$, $P_{G}$ and $P_{B}$, respectably enabling to determine the vectors ${\vec{r}}_{BR}$ and ${\vec{r}}_{BG}$. To calculate the deformation angle α we use the Equation (Equation 1).
(1)$$\begin{array}{c} {\vec{r}}_{BR}\cdot {\vec{r}}_{BG}=|{\vec{r}}_{BR}||{\vec{r}}_{BG}|\cos\left(\alpha \right). \end{array}\;$$

The equation to calculate α is
(2)$$\begin{array}{c} \alpha =\arccos\left(\frac{{\vec{r}}_{BR}\cdot {\vec{r}}_{BG}}{\left|{\vec{r}}_{BR}||{\vec{r}}_{BG}\right|}\right). \end{array}\;$$

The deformation angle α is calculated in radians. For the experiments, a conversion was performed to present the results in degrees.

The system works as follows. The operation is initiated when the actuator starts to deflect. The camera captures the deformation as an image and then transmits it to MATLAB for processing. In MATLAB, the image is analyzed to calculate the deformation angle, as previously described. Also in MATLAB, the input signal and control parameters are configured to generate the control signal sent to the Arduino producing a PWM (Pulse-width modulation) signal sent to the driver circuit. The driver circuit supplies the appropriate current to heat the SMA wire, achieving the desired deformation angle. This closed loop is run in real-time with a period of 0.025s. The data flow described is shown in Figure 5.

## 4. Modeling

A mathematical model was developed to describe the dynamic behavior of the actuator system using system identification, which involves measuring the inputs and outputs related to the system and constructing a model that aligns closely with the measured data. The soft actuator exhibits bidirectional movement. In this study, when SMA wire 1 is activated, it induces a clockwise bending, measured as a negative deformation. Conversely, when voltage is applied to SMA wire 2, it results in counterclockwise bending of the actuator, measured as a positive deformation. This behavior is illustrated in Figure 6.

A methodology similar to the one employed in [19,20] was applied to develop the model. Initially, multiple actuation tests were conducted to obtain the data for both input and output variables. Figure 7 illustrates the applied input and the corresponding outputs obtained from the experiments. When SMA wire 1 is heated, the actuator achieves a maximum deformation angle of 31.7°. Similarly, heating the second SMA wire results in a maximum deformation angle of 30°. The inability of the actuator to return to the neutral position of 0° during the cooling period from 25 s to 50 s, precisely illustrates the necessity for the development of a control system. Therefore, it is important to clarify that in this phase, the behavior of the actuator is not actively controlled or influenced by feedback. Rather, it represents a direct measure of the natural heating and subsequent cooling of the actuator in response to the applied voltage on the SMA wires. The observed behavior is indicative of the inherent characteristics and hysteresis of the actuator.

The observed output behavior obtained in the experiments aligns with a first-order linear time-invariant system, exhibiting no overshoot. Subsequently, a system identification process is employed to derive the parameters of a first-order transfer function, as outlined in Equation (Equation 3), where *K* represents the gain and *T* denotes the time constant. This identification process utilizes the input and output data stored for each experiment, this data helps to estimate the parameters of the transfer function model. The model is formulated to represent a first-order system, and the resulting numerator and denominator coefficients are used to construct the transfer function *G*.
(3)$$\begin{array}{c} G\left(s\right)=\frac{K}{1+sT}. \end{array}\;$$

Subsequently, the identified model is simulated in Simulink to facilitate a comparison between the experimental data and the simulated results. This iterative procedure enables a comprehensive evaluation and validation of the model using experimental datasets, ensuring precision and consistency. An illustrative example of such a comparison is presented in Figure 8.

For each experiment, a transfer function was computed as shown in Equation (Equation 4) where *j* denotes the number of the experiment.
(4)$$\begin{array}{c} G_{j}\left(s\right)=\frac{K_{j}}{1+sT_{j}}.\; \end{array}$$

The parameters were computed for each experiment, resulting in a dataset for each parameter. Based on the experimental results, we determined the range of the gain *K*, measured in degrees per volt, and the range of the time constant *T*, measured in seconds. In Equations (Equation 5) and (Equation 6) the upper and lower values for each parameter are shown.
(5)$$\begin{array}{c} K\in \left[\underline{K},\bar{K}\right]=\left[4.402\;{}^{\circ}/V,\;4.509\;{}^{\circ}/V\right]. \end{array}$$
(6)$$\begin{array}{c} T\in \left[\underline{T},\bar{T}\right]=\left[6.570\;s,\;7.474\;s\right].\; \end{array}$$

The nominal transfer function of the system $\tilde{G}\left(s\right)$ is calculated, which corresponds to the average between the maximum and minimum values for each parameter. Therefore, the nominal transfer function is determined as,
(7)$$\begin{array}{c} \tilde{G}\left(s\right)=\frac{\tilde{K}}{1+s\tilde{T}}=\frac{4.456}{1+7.022s}.\; \end{array}$$

It is important to mention that these sets of parameters are bounded between lower and upper values as can be seen in the frequency responses of the identified first-order models that are shown in Figure 9, in this case, the nominal transfer function is located in the middle of the other obtained models. In comparison with the results obtained in [10], where the SMA wires were overbraided with copper wires and polyamide yarns, the SMA wires are embedded in PTFE tubes. As a result of this manufacturing change, the data obtained from the experiments exhibit less uncertainty in the parameters, especially in the parameter *T*. This reduction in uncertainty leads to smaller variations and improved repeatability in experiments.

## 5. Control

A fundamental aspect of designing a robust controller is considering potential uncertainties that could impact the behavior of the actuator. These uncertainties might include unknown nonlinearities, parametric variations, non-modeled dynamics, and responses to certain external forces such as gravity, among others. Based on [27] and the research conducted in [19,20], a robust stability approach is applied to achieve a good performance of the prototype, even in the face of uncertainties. In this context, additive and multiplicative uncertainty models were used, assuming that the nominal transfer function of the plant is denoted by $\tilde{G}\left(s\right)$. The perturbed transfer function G(s) can be described for additive uncertainty as the Equation (Equation 8) and for multiplicative uncertainty as the Equation (Equation 9).
(8)$$\begin{array}{cc} G(s) & =\tilde{G}\left(s\right)+W\left(s\right)\Delta \left(s\right). \end{array}$$
(9)$$\begin{array}{cc} G(s) & =\tilde{G}\left(s\right)\left(1+W\left(s\right)\Delta \left(s\right)\right).\; \end{array}$$

In the previous equations, W(s) is a proper and stable weight function that characterizes the uncertainty dynamics. Meanwhile, Δ(s) contains the uncertainty itself, which can be represented by any stable transfer function that satisfies the following inequality.
(10)$$\begin{array}{c} {\left||\Delta |\right|}_{\infty }\leq 1. \end{array}\;$$

To meet these conditions, suitable weighting functions W(s) and Δ(s) for additive and multiplicative uncertainty, respectively, were derived based on the findings in [19]. These are presented in Equations (Equation 11) and (Equation 12) for additive uncertainty.
(11)$$\begin{array}{c} W\left(s\right)=\frac{s\left(\bar{K}\tilde{T}-\tilde{K}\underline{T}\right)+\bar{K}-\tilde{K}}{s^{2}\tilde{T}\underline{T}+s\left(\tilde{T}+\underline{T}\right)+1}. \end{array}\;$$
(12)$$\begin{array}{c} \Delta \left(s\right)=\left(\frac{s^{2}\tilde{T}\underline{T}+s\left(\tilde{T}+\underline{T}\right)+1}{s^{2}\tilde{T}T+s\left(\tilde{T}+T\right)+1}\right)\left(\frac{s\left(K\tilde{T}-\tilde{K}T\right)+K-\tilde{K}}{s\left(\bar{K}\tilde{T}-\tilde{K}\underline{T}\right)+\bar{K}-\tilde{K}}\right). \end{array}$$
For multiplicative uncertainty in Equations (Equation 13) and (Equation 14).
(13)$$\begin{array}{c} W\left(s\right)=\frac{s\left(\bar{K}\tilde{T}-\tilde{K}\underline{T}\right)+\bar{K}-\tilde{K}}{\tilde{K}\left(s\underline{T}+1\right)}. \end{array}$$
(14)$$\begin{array}{c} \Delta \left(s\right)=\left(\frac{s\left(K\tilde{T}-\tilde{K}T\right)+K-\tilde{K}}{s\left(\bar{K}\tilde{T}-\tilde{K}\underline{T}\right)+\bar{K}-\tilde{K}}\right). \end{array}$$

For additive uncertainty, according to [27], the closed-loop system is robustly stable only if inequality (Equation 15) is satisfied.
(15)$$\begin{array}{c} ||WC\tilde{S}{\left|\right|}_{\infty }<1. \end{array}\;$$
In the case of the multiplicative uncertainty, the condition for robust stability is given by the inequality (Equation 15) and (Equation 16):(16)$$\begin{array}{c} ||W\tilde{T}{\left|\right|}_{\infty }<1. \end{array}$$

To regulate the deformation angle of the soft actuator, a proportional-integral (PI) controller is proposed, specifically because the system is a first-order system. The controller gains $K_{p}$ and $K_{i}$ are tuning according to robust stability conditions. The transfer function for the control is shown in the Equation (Equation 17), where C(s) is the controller, U(s) is the control signal, and E(s) is the error.
(17)$$\begin{array}{c} C\left(s\right)=\frac{U(s)}{E(s)}=K_{p}+\frac{K_{i}}{s}. \end{array}$$

The (nominal) open-loop transfer function L(s) of the system with the controller is expressed as:(18)$$\begin{array}{c} L\left(s\right)=\left(K_{p}+\frac{K_{i}}{s}\right)\left(\frac{\tilde{K}}{1+s\tilde{T}}\right). \end{array}$$

The (nominal) corresponding complementary sensitivity functions $\tilde{S}\left(s\right)$ and $\tilde{T}\left(s\right)$ are:(19)$$\begin{array}{c} \tilde{S}\left(s\right)=\frac{s(1+s\tilde{T})}{s^{2}\tilde{T}+s\left(\tilde{K}K_{p}+1\right)+\tilde{K}K_{i}}. \end{array}\;$$
(20)$$\begin{array}{c} \tilde{T}\left(s\right)=\frac{\tilde{K}\left(sK_{p}+K_{i}\right)}{s^{2}\tilde{T}+s\left(\tilde{K}K_{p}+1\right)+\tilde{K}K_{i}}. \end{array}\;$$

By substituting the provided equations into the robust stability condition for each uncertainty model, a range of valid and invalid combinations for $K_{p}$ and $K_{i}$ is calculated to ensure the robust stability of the system. The graphical representation of these combinations can be observed in Figure 10.

## 6. Experimental Results

The controller in the prototype aims to regulate the deformation angle of the soft actuator. Using the test bench with the computer vision system, the closed-loop control diagram illustrated in Figure 11 was implemented in Matlab. Where $X_{d}$ is the desired deformation angle, E(s) is the error, C(s) is the controller, U(s) is the control signal, G(s) is the actual prototype, and X(s) is the actual deformation angle. The gains of the PI controller are $K_{p}=50$ and $K_{i}=3.2$, which satisfy the robust stability conditions.

Figure 12 illustrates the reference $x_{d}$ in red and the system output *x* in blue during an experiment. The reference consists of multiple steps, ranging from 0° to 20° in both directions. As shown in the figure, the actuator successfully reaches the reference with a precise and stable response. The performance of the controller can be quantified by observing the error signal, shown in Figure 13, the graph indicates that the error in steady state for each step is bounded within the region |e| < 1°.

Figure 14 shows the behavior of the system control signal. It illustrates the voltage required to achieve or maintain specific references. As can be seen, the control signal is bounded and is equal to or less than the voltage supplied by the power source, which is 6 V.

## 7. Conclusions

In this study, we improved the design and controller of a soft actuator capable of bidirectional deformation. The manufacturing process involved encasing SMA wires within PTFE tubes, which significantly reduced uncertainties in the system. Additionally, a computer vision system was implemented to measure angle deformations, enhancing the mathematical model by providing detailed and reliable data. This improvement led to reduced uncertainties in the parameters and better performance in the control design.

A robust control strategy was implemented using PI control, ensuring precise actuation to achieve specific deformation angles. The performance of the controller was validated through experiments, demonstrating accurate control in reaching desired deformations with an error margin within ±1° for the steady state.

Robustness conditions within the controller design aim to ensure the ability of the soft actuator to withstand variations in the structure and disturbances. However, the current study does not include experiments specifically addressing the effects of increased actuation cycles on the actuator. Recognizing the value of such investigations for a deeper understanding of the prototype, future studies will be conducted to analyze and perform these experiments. The goal is to gain insights into structural changes over prolonged actuation cycles and to refine the model as well as control strategies accordingly.

For future work, we plan to continue using 3D manufacturing techniques, enabling us to explore diverse structures and designs. These advancements, including 3D manufacturing methods, sensing techniques, modeling methodologies, and robust controllers, will drive our future research efforts, enhancing the development of innovative soft actuators that could be used in different areas of study.

## Figures and Tables

**Figure 1 materials-17-00758-f001:**
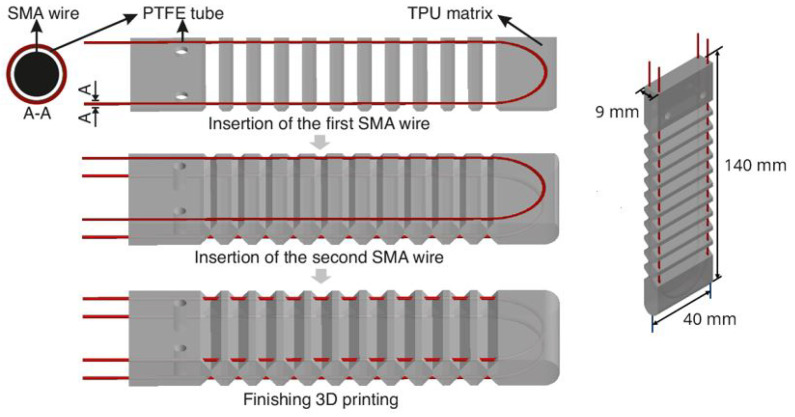
Two-direction soft actuator manufacture process.

**Figure 2 materials-17-00758-f002:**
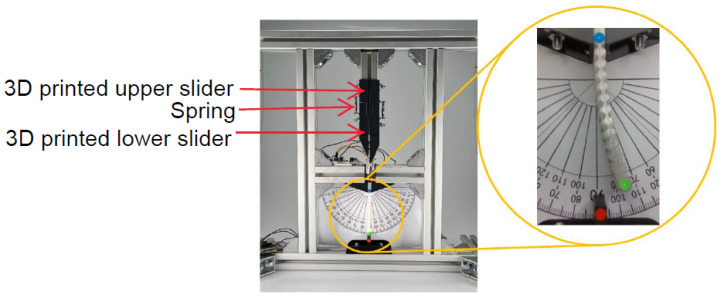
Photographs of the test bench and the actuator.

**Figure 3 materials-17-00758-f003:**
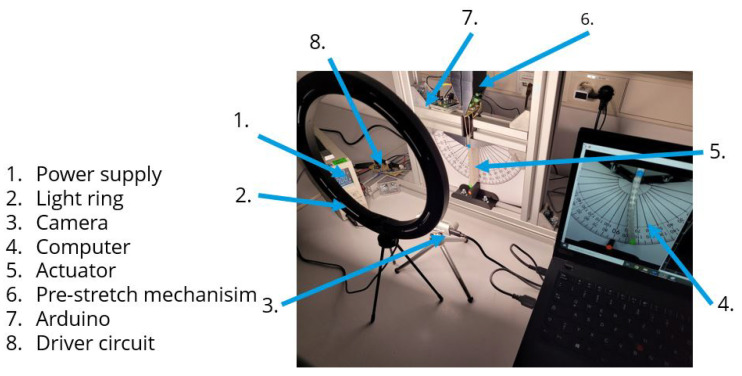
Experimental setup for the computer vision system.

**Figure 4 materials-17-00758-f004:**
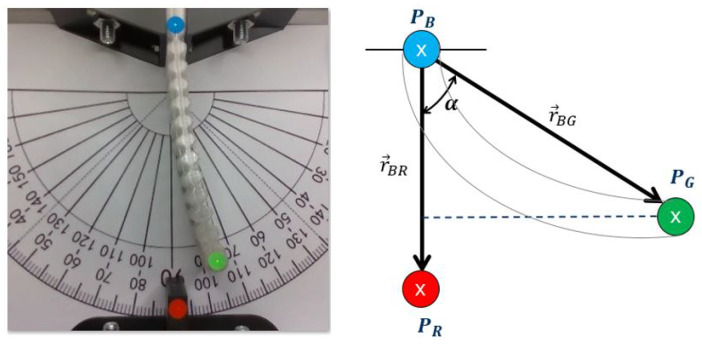
Photograph of the actual actuator (**left**), schematic diagram of the variables used to calculate the deformation angle α (**right**).

**Figure 5 materials-17-00758-f005:**
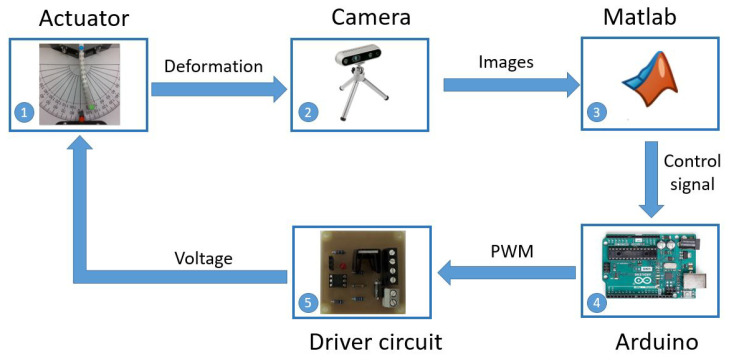
Data flow diagram of the computer vision system.

**Figure 6 materials-17-00758-f006:**
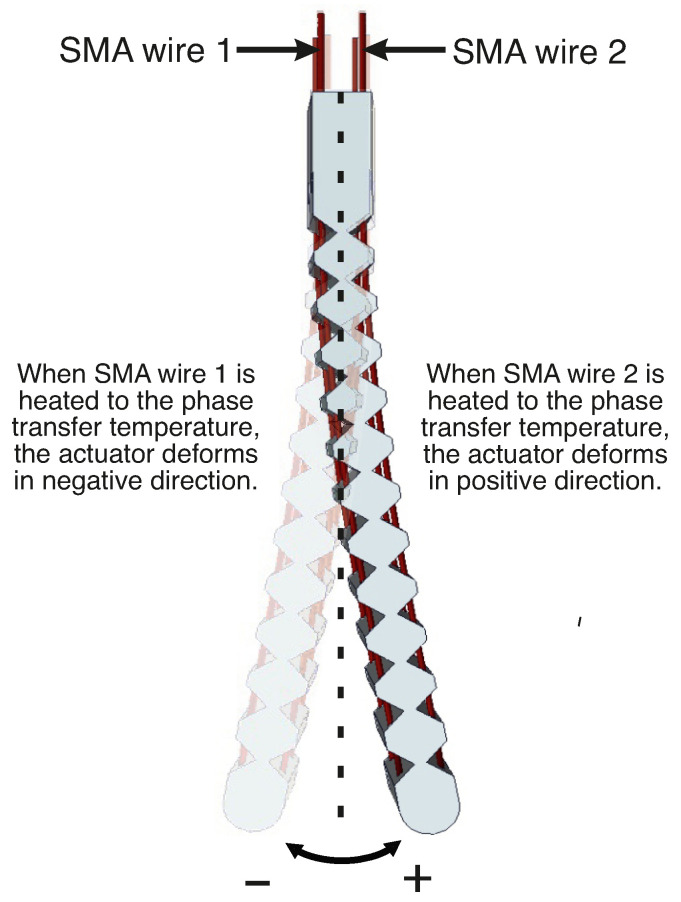
Actuator activation mechanism and definition of the bending directions.

**Figure 7 materials-17-00758-f007:**
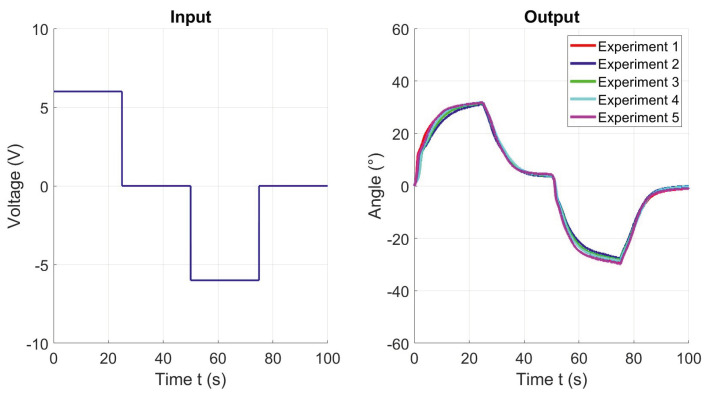
Input and output data of 5 experiments for system identification.

**Figure 8 materials-17-00758-f008:**
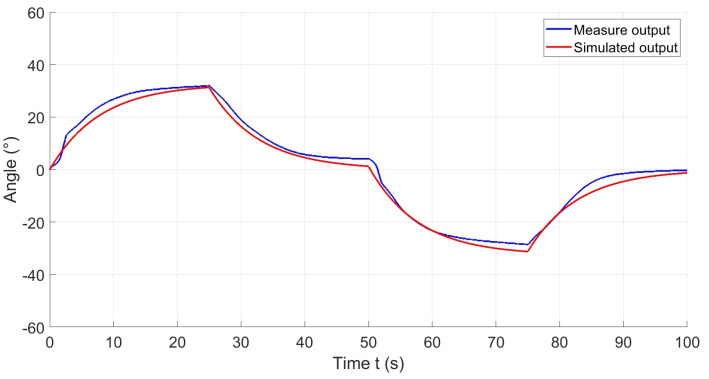
Comparison of the experimental output with the simulated output.

**Figure 9 materials-17-00758-f009:**
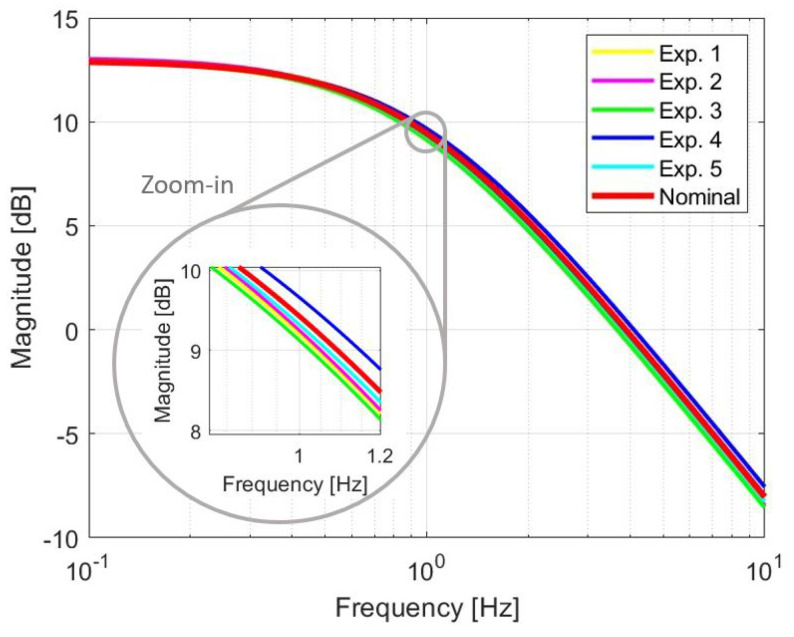
Frequency responses of the identified first-order models.

**Figure 10 materials-17-00758-f010:**
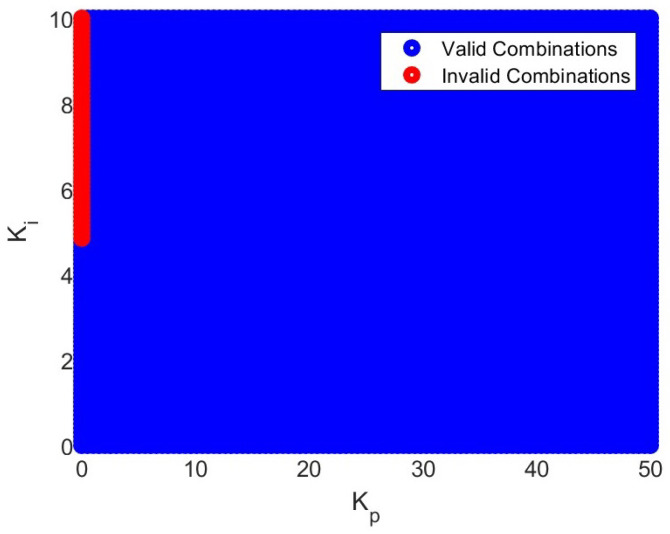
Combination values for $K_{p}$ and $K_{i}$ to satisfy robust stability condition for additive and multiplicative uncertainty models.

**Figure 11 materials-17-00758-f011:**
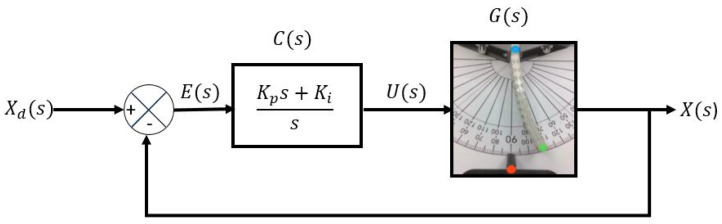
Closed-loop diagram.

**Figure 12 materials-17-00758-f012:**
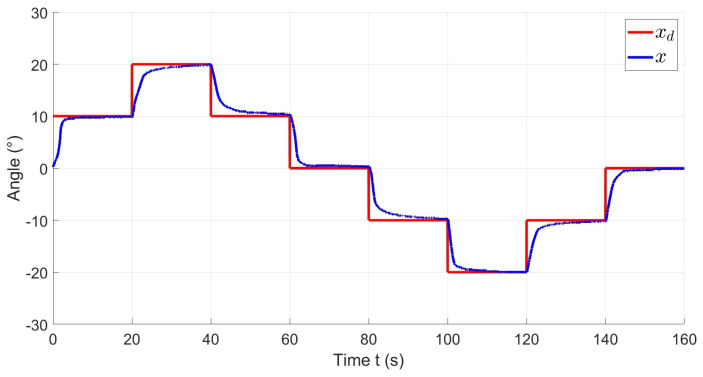
Desired deformation angle $x_{d}$ and actual deformation angle *x*.

**Figure 13 materials-17-00758-f013:**
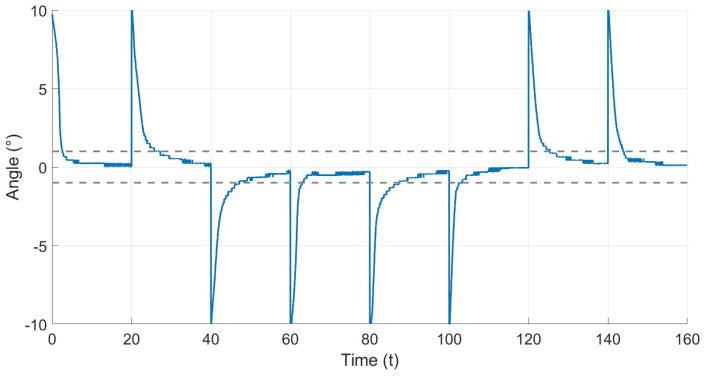
Error of the PI controller.

**Figure 14 materials-17-00758-f014:**
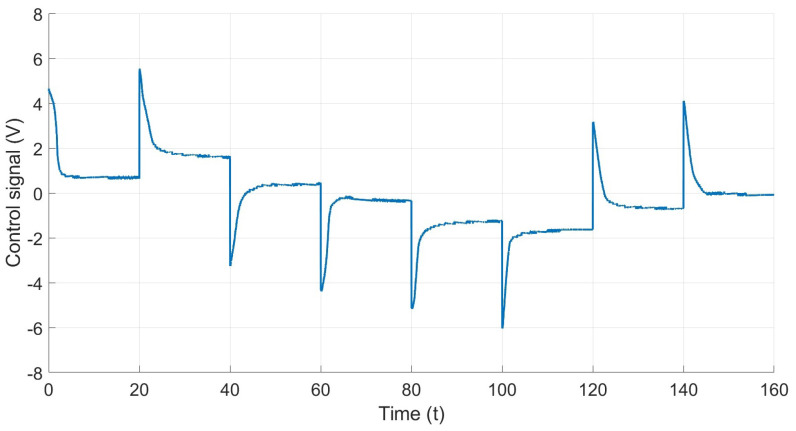
Control signal of the PI controller.

## Data Availability

Data are contained within the article.

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
