# Peer review of "Manufacture and Deformation Angle Control of a Two-Direction Soft Actuator Integrated with SMAs"

_materials, 2024, doi:10.3390/ma17030758_

Round 1
Reviewer 1 Report
Comments and Suggestions for Authors
page 1- discuss details of 3D actuator - discuss the actuator
several companies are involved - how assesss?
how is done fig. 6? what is G(s) - what is K E (cannot assess simbols in fig. 7, 8. discuss further)
Comments on the Quality of English LanguageThe descrption should be mostly filled of details in order to understand
the MatLAB script should be shown. Page 6 should very difficult to unedrstand; for example: G(s) = K/1+sT?
Finally, the notation in [149 to 151] is not understsand
Reviewer 2 Report
Comments and Suggestions for Authors
Dear Authors,
you can find the reviewer's comments in the file attached.

Moderate editing of the English language is required.
Reviewer 3 Report
Comments and Suggestions for Authors
The manuscript presents a soft actuator integrated with SMA wires that is able to bend in two directions. 3D printing was adopted to fabricate the main body of the actuator, while SMA-embedded PTFE tubes were placed in position relying on ‘pause at height’ method. A computer vision system was built to monitor the bending angles. Bending results were utilized to establish a model, which was verified by capturing the experiment results well. A control strategy considering additive uncertainty or multiplicative uncertainty was developed, ensuring precise actuation control. On the whole the manuscript was in good shape. One minor issue that needs attention is Equation 6, where the values in the numerator and denominator should be switched. And to make the description more rigorous, the authors need to consider:
1. How did the authors consider the impact of gravity on the nonlinear bending deformation?
2. How did the authors decide the steady deformation angle after applying a voltage? Rationale?
3. In Fig 7, why could the actuator not go back to neutral 0 deg status in the second phase [t: 25-50 s]?
Reviewer 4 Report
Comments and Suggestions for Authors
In this manuscript, A. I. Acevedo-Velazquez, et al developed an innovative two-direction soft actuator based on 3D printed TPU composite with embedded shape memory alloys (SMA) wires. The experimentally obtained parameters of the actuator from the computer vision system were used to develop a mathematical model. The results show that the developed actuator and the associated controlling system provide good deformation performance with lower uncertainty and improved repeatability. The presented detailed experimental results and analysis method are reasonable and interesting, which may provide important contributions to the application of SMA wires in developing 3D-printed soft actuators. Therefore, I suggest the publication of this manuscript in Materials with minor revisions.
1) What is the thickness of the PTFE tube? Will a change in the wall thickness influence the performance of the actuator?
2) Since the parameters of the actuator are experimentally measured, will increasing the actuation cycle change the parameters of the system due to the permanent deformation of the sturctures? This may influence the accuracy of the developed controlling system. Therefore, the author should discuss this in the manuscript.
Round 2
Reviewer 1 Report
Comments and Suggestions for Authors
The new discussion has been stregnthed, such that I approve the paper.